# FAST-FUSION: An Improved Accuracy Omnidirectional Visual Odometry System with Sensor Fusion and GPU Optimization for Embedded Low Cost Hardware †

**André Aguiar** [1,*], **Filipe Santos** [1], **Armando Jorge Sousa** [1,2] and **Luís Santos** [1]

1 INESC TEC—INESC Technology and Science; 4200-465 Porto, Portugal; fbsantos@inesctec.pt (F.S.); luis.c.santos@inesctec.pt (L.S.)
2 Faculty of Engineering, University of Porto; 4200-465 Porto, Portugal; asousa@fe.up.pt
\* Correspondence: andre.s.aguiar@inesctec.pt
† This paper is an extended version of our paper published in ROBOT'2019.

**Abstract:** The main task while developing a mobile robot is to achieve accurate and robust navigation in a given environment. To achieve such a goal, the ability of the robot to localize itself is crucial. In outdoor, namely agricultural environments, this task becomes a real challenge because odometry is not always usable and global navigation satellite systems (GNSS) signals are blocked or significantly degraded. To answer this challenge, this work presents a solution for outdoor localization based on an omnidirectional visual odometry technique fused with a gyroscope and a low cost planar light detection and ranging (LIDAR), that is optimized to run in a low cost graphical processing unit (GPU). This solution, named FAST-FUSION, proposes to the scientific community three core contributions. The first contribution is an extension to the state-of-the-art monocular visual odometry (Libviso2) to work with omnidirectional cameras and single axis gyro to increase the system accuracy. The second contribution, it is an algorithm that considers low cost LIDAR data to estimate the motion scale and solve the limitations of monocular visual odometer systems. Finally, we propose an heterogeneous computing optimization that considers a Raspberry Pi GPU to improve the visual odometry runtime performance in low cost platforms. To test and evaluate FAST-FUSION, we created three open-source datasets in an outdoor environment. Results shows that FAST-FUSION is acceptable to run in real-time in low cost hardware and that outperforms the original Libviso2 approach in terms of time performance and motion estimation accuracy.

**Keywords:** mobile robots; visual odometry; sensor fusion; heterogeneous computing

## 1. Introduction

The main task while developing a mobile robot is to achieve secure and robust navigation in a given environment. The environment defines the type of navigation, i.e., a mobile robot can perform indoor, outdoor in a structured environment or outdoor in an unstructured environment navigation [1]. To achieve such a goal, the ability of the robot to localize itself is crucial. In outdoor environments, this task becomes a real challenge. The higher density of moving objects, the terrain irregularities, and the characteristics of illumination that are present in an outdoor environment make robot motion estimation difficult and, consequently, the process of localizing it [2]. In such conditions, sensors like inertial measurement units (IMU) or encoders tend to present considerable errors. One of the most common solutions is to use satellite-based localization systems by using a receiver of global navigation satellite systems (GNSS). However, in confined environments such as tunnels, urban canyons, or steep

slope hills, a signal blockage might occur, which makes the use of GNSS an unreliable solution [3]. Hereupon, the development and use of a satellite redundant localization system are essential. In this context emerges visual odometry (VO). In 2004, Nister et al. [4] created this concept based on the fundamentals of wheel odometry. VO in the monocular case uses a single camera to track the robot motion between consecutive image frames [5]. On one hand, this consists of a hardware-inexpensive solution. On the other hand, the use of a single camera has inherent issues. Estimating the relative motion of the robot while it is performing pure rotations can lead to high errors due to the fast change of the world view and consequent low overlap between images. Also, to perceive the motion scale of a mobile robot using a single camera without any prior knowledge about the environment or other sources of information is not possible due to the unavailability of depth information [6]. To improve VO performance, the use of omnidirectional cameras in this context has increased since they allow to capture more information about the scene and to track individual features over a more extensive set of consecutive images [7]. Even so, monocular omnidirectional VO benefits from the use of other sources of information such as inertial and/or range sensors. With a fusion of information from a variety of sources, it is possible to extract the main features of each one and deal more robustly with their limitations.

To obtain an acceptable motion estimation using a single camera and custom sensors can be time intensive. To achieve real-time performance under these conditions, optimization techniques should be used. In this context, the migration from homogeneous to heterogeneous computing is a logical solution. Heterogeneous computing was defined as "the well-orchestrated and coordinated effective use of a suite of diverse high-performance machines (including parallel machines) to provide superspeed processing for computationally demanding tasks with diverse computing needs" [8]. One of the standard topologies used in heterogeneous computing is a combination of core processing unit- (CPU) and graphical processing unit- (GPU)-based resources. The collaboration between these two processing units is a key approach to achieve high levels of performance in nowadays systems [9]. Usually, the CPU works as host where the main code is executed while the GPU runs the so-called kernels. To facilitate the implementation of this topology, open computing language (OpenCL) [10] was developed. This framework supports a variety of devices such as CPUs, GPUs, and others. It supports C-based programming languages to work on these devices and offers application programming interfaces (API) to facilitate the interface between the mentioned devices.

In this work, we propose a localization system suitable for ground robots with GPU-based optimization techniques, which we call FAST-FUSION. The central module of our system is a version of Libviso2 [11] for omnidirectional cameras that is publicly available at the original Libviso2 repository (https://github.com/srv/viso2). To improve the VO method in rotational motions, we propose a Kalman filter (KF) to fuse the orientation from VO with a gyroscope. Besides estimating the angular velocity, we also estimate the gyroscope bias online. To estimate the motion scale, we use a planar light detection and ranging (LIDAR) sensor. The system runs and is validated on a Raspberry Pi 3B where we use both its CPU and GPU with OpenCL-based optimizations.

The rest of the paper is described as follows. In the next section the related work is presented. Section 3 describes an overview of the system architecture. Section 4 contains the approach adopted in this work. In particular, the omnidirectional VO method, the fusion of both the gyroscope and the LIDAR with it, and the GPU-based optimizations. Section 5 exposes the test results of the system using built in-house datasets. Section 6 presents the discussion of the obtained results. Finally, the work is summarized in Section 7.

## 2. Related Works

This section describes the current state-of-art of the omnidirectional VO field, as well as the approaches to solve its main vulnerabilities such as sensor fusion. Also, parallel computing approaches in this field are briefly reviewed.

### 2.1. Omnidirectional Visual Odometry

To improve the motion estimation accuracy in the VO context, the extraction of reliable information about the world scene is crucial. This way, the use of omnidirectional cameras in this field has become more and more common. If a raw omnidirectional image stream is used, on one hand, a deeper knowledge about the environment is obtained, but on the other hand, it is required to deal with the distortion present in these images.

Many works that use omnidirectional cameras in VO have been proposed in the literature. Some state-of-the-art original VO methods were extended to the use of these devices. For example, direct sparse odometry (DSO) [12] was adapted in this way. The omnidirectional version [13] uses a full image area even in the presence of strong distortion. It takes advantage of the high number of point overlap between images due to the higher field of view. In the same way, LSD-SLAM [14] was extended. The new version [15] uses raw omnidirectional images to extract more information about the world, which results in a performance improvement even in pure rotations. They use a 185º fisheye lens to test and evaluate their work. Zhang et al. propose a version of semi-direct visual odometry (SVO) [16] that also uses wide field of view cameras. They implement a camera model suitable for these cameras and use samples from the epipolar curves present in the omnidirectional VO configuration to estimate the camera motion [7].

Some built from origin omnidirectional VO methods are also present in the literature. For example, Corke et al. presented a catadioptric camera configuration for planetary rovers in GPS-denied environments [17]. In this work, two approaches are proposed: the first is optical-flow-based; the second is structure-for-motion-based. To evaluate their approach, a sequence with 2000 images is used. Similarly, two different works present the same configuration [18,19]. The first is a monocular VO system that uses an omnidirectional camera placed on top of a car to estimate motion in outdoor environments. This system presents two different trackers—a feature-based that uses scale-invariant feature transform (SIFT) features and an appearance-based that works as a visual compass. In the second, they use four cameras with aligned optical centers to simulate an omnidirectional one. The rotation and translation components were decoupled to estimate the robot pose. This approach reaches one of the longest paths (2.5 km) reported in VO history with an high level of accuracy. Valiente et al. present two different systems that use an omnidirectional camera [20,21]. The first uses an omnidirectional camera in a VO method to generate a reliable input for a mapping approach. The last presents several contributions, such as a strategy to deal with the scale uncertainty present in the monocular scheme and an epipolar constraint adapted to the omnidirectional geometry. Finally, Li et al. present a system that works with full-view omnidirectional images that are used in a spherical-model-based simultaneous localization and mapping (SLAM) approach in an indoor environment [22].

### 2.2. Visual Odometry Challenges

Developing a standalone monocular VO system is a challenging task due to some limitations present in this approach. Using a single camera to track the motion of a robot has two main problems:

- Scale ambiguity due to the incapacity to perceive the scene depth without any prior information about the environment or an additional source of information.
- Estimation degeneration on pure rotations due to the low overlap between consecutive images.

In this context, the use of sensors to support VO is important. Inertial sensors are widely used to do so as they can resolve the ambiguities imposed by the monocular scheme [23]. They can be used to refine the estimation of the rotational component [24], to improve the visual estimator performance using short-term motion constraints [25] and/or to be used alongside VO to improve the general motion estimation [26]. Besides this, inertial sensors can also be used to recover the motion scale [27]. However, to perform this task, two main approaches are usually adopted. The first is to use some prior knowledge about the environment, such as the camera height and/or pitch [6,11], or the size of some

known object present in the camera field of view [28]. The second is the use of range sensors [29–31], which usually relates the distance measures with the camera perception to recover the depth of a set of pixels.

Some VO approaches can also be computationally expensive. Many of them use iterative methods that can difficult their use in real-time embedded systems. A solution to speed up these methods is to use heterogeneous computing. There are a few approaches that use GPU-based optimizations to do so. For example, Zhang et al. propose a CUDA acceleration for a robot localization and mapping approach [32] where a NVIDIA GPU is considered to run a particle filter. Similarly, Delgado et al. [33] propose a VO-optimized approach with GPU-OpenCV optimizations.

## 3. FAST-FUSION System Architecture

The proposed system aims to localize in real-time a ground robot in an agricultural environment. To do so, we propose a fusion of sensors with a monocular omnidirectional VO algorithm. The entire system runs on top of a low-cost microprocessor, a Raspberry Pi 3B. An OpenCL-based optimization approach applied to the VO method is proposed recurring to the Raspberry Pi's GPU to overcome the limitations of this microprocessor. The system is summarized in Figure 1.

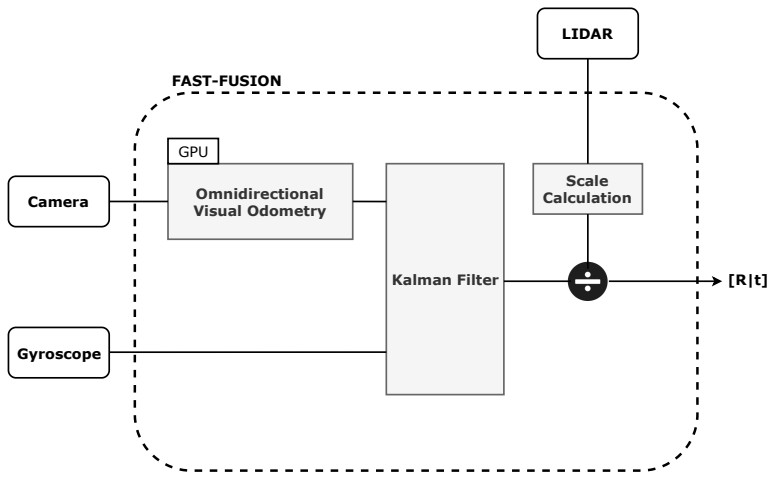

**Figure 1.** High-level system architecture.

The central unity of the system is the VO method. This one is publicly available on the official Libviso2 repository and can work standalone, giving a primary estimation of the robot motion. A sensory system is also proposed to support and solve the main limitations of it. As can be observed, the sensory system is constituted by a planar laser and a gyroscope. The first is used to calculate the motion scale due to the unavailability of depth information resultant from a monocular VO system. The last is used as a support to VO in rotations and it is fused with VO recurring to a KF. The system output is a homogeneous transformation $[\mathbf{R}|\mathbf{t}]$ between consecutive image frames.

## 4. FAST-FUSION Approach

### 4.1. Omnidirectional Visual Odometry

As referenced before, the central module of our system is an omnidirectional VO method that is an extension of the state-of-the-art VO approach Libviso2 [11]. Figure 2 is a high-level representation of our approach. As can be observed, it is divided into three main steps:

1. Application of a camera model that converts 2-D feature pixels in the omnidirectional image in 3-D unit vectors.
2. A Random sample consensus (RANSAC) approach to select the inliers from the entire set of 3-D unit vectors.

3.    Motion estimation using the epipolar constraint and linear triangulation.

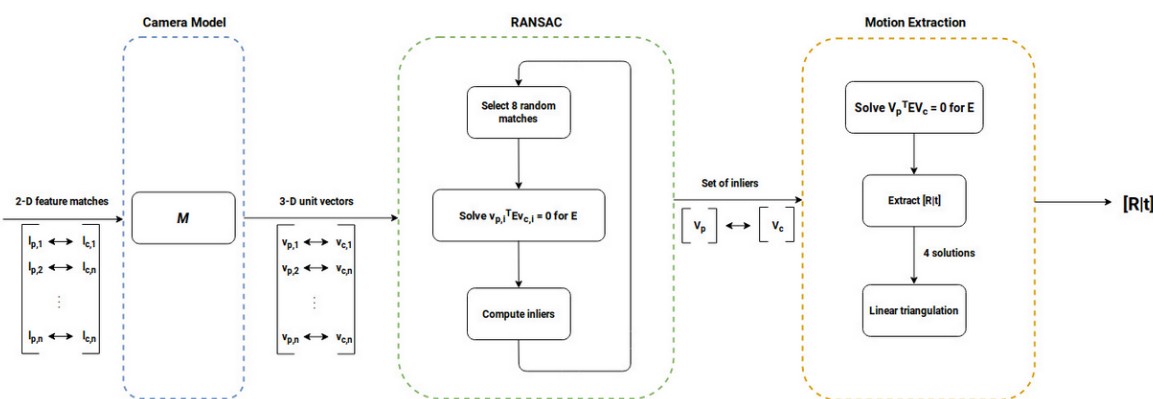

**Figure 2.** Omnidirectional visual odometry scheme.

To deal with the distortion imposed by an omnidirectional image, a suitable camera model is required. We chose the unified camera model [34] proposed by Davide Scaramuzza et al. This model uses the calibration parameters obtained from the Matlab toolbox [35] provided by the authors. Figure 3 represents the two transformations provided by this model. It allows us to convert a 2-D pixel in a 3-D unit vector but also to use the inverse model to transform a world point in a 2-D pixel.

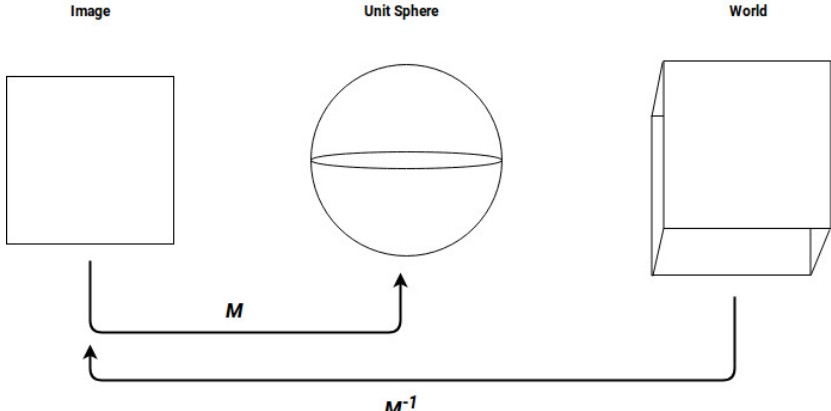

**Figure 3.** Camera model overview.

The camera model is represented by a polynomial. The polynomial and the inverse polynomial coefficients are obtained during the calibration procedure. This being said, the model is described as follows.

**Definition 1.** *Let* $X = [x \ y \ z]^T$ *be a scene point observed by the omnidirectional camera,* $x'' = [u'' \ v'']^T$ *its projection into the sensor plane,* $x' = [u' \ v']^T$ *in the camera plane and $v$ be the unit vector that emanates from the viewpoint to the scene point. The camera plane refers to the image plane and it is expressed in pixels. The sensor plane is an hypothetical plane orthogonal to the mirror axis or fish-eye lens, with the origin located at the camera optical center expressed in metric coordinates, i.e., pixels in relation with the image center. The projection of a point in the camera plane into the unit sphere is given by:*

$$\frac{\left[u' \ v' \ f(u',v')\right]^T}{\left\| \left[u' \ v' \ f(u',v')\right]^T \right\|}, \ f(u',v') = a_0 + a_1 r' + ... + a_N r'^N \tag{1}$$

where $x' = A^{-1}(x'' - t)$ is the affine transformation that converts points in the sensor plane into the camera plane and $f(u', v')$ is the polynomial function that represents the mirror/lens distortion and gives information about the direction of the 3-D ray that emanates from the viewpoint to the 3-D scene point in function of the euclidean distance $r' = \sqrt{u'^2 + v'^2}$ of the image point to its respective center.

**Definition 2.** *Let $X = [x\ y\ z]^T$ be a scene point observed by the omnidirectional camera where $z$ represents its depth and $h(u', v')$ the inverse polynomial of $f(u', v')$. To project this 3-D point into the image the following transformation is performed*

$$x'' = \begin{bmatrix} x\frac{\theta h_1 + \theta^2 h_2 + ... + \theta^N h_N}{\sqrt{x^2+y^2}} \\ y\frac{\theta h_1 + \theta^2 h_2 + ... + \theta^N h_N}{\sqrt{x^2+y^2}} \end{bmatrix} A + \begin{bmatrix} x_c \\ y_c \end{bmatrix}, \ \theta = tan(\frac{z}{\sqrt{x^2 + y^2}}) \tag{2}$$

*where $h_i$ is the ith coefficient of the inverse polynomial $h(u', v')$, $A$ is the affine matrix of the camera model and $[x_c\ y_c]^T$ is the image center.*

After having a model $M$ that converts the 2-D features in 3-D unit vectors dealing with the camera distortion, we propose an adaptation of Libviso2 to estimate the camera motion. To do so, we reuse the matching procedure from the original approach and recreate the epipolar geometry approach to work with omnidirectional cameras. Starting with a set of 2-D feature matches $(\mu_c, \mu_p)$ between the previous $p$ and current $c$ images $\{\mu_p\} \longleftrightarrow \{\mu_c\}$ with $\mu_p = \{x''_{p_1}, x''_{p_2}, ..., x''_{p_n}\}$ and $\mu_c = \{x''_{c_1}, x''_{c_2}, ..., x''_{c_n}\}$ we apply $M$ to them obtaining a set of 3-D unit vector matches $\{\eta_p\} \longleftrightarrow \{\eta_c\}$ with $\eta_p = \{v_{p_1}, v_{p_2}, ..., v_{p_n}\}$ and $\eta_c = \{v_{c_1}, v_{c_2}, ..., v_{c_n}\}$. Since we use 3-D unit vectors, the conventional epipolar geometry configuration is not suitable. This being said, we use the configuration present in Figure 4.

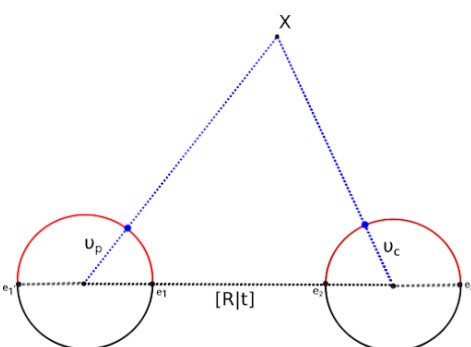

**Figure 4.** Epipolar geometry configuration.

In this, instead of projecting the line that contains the previous camera center and the scene point into the current image plane—the so-called epipolar line— we project it into the current unit sphere. With this configuration, we have now epipolar curves (represented in red in Figure 4) instead of epipolar lines. This means that a point on the unit sphere correspondent to the current image that matches a point on the unit sphere correspondent to the last image lies on an epipolar curve. With this, we have the RANSAC input prepared—the entire set of 3-D unit vector matches—and we are able to compute this method solving for the essential matrix $E$. So for each iteration, we select a random sample of size eight from the total set of unit vectors and solve

$$v_{p_i} E v_{c_i} = 0 \tag{3}$$

for each one. So, for each match we have

$$\begin{bmatrix} x_{p_i} & y_{p_i} & z_{p_i} \end{bmatrix} E \begin{bmatrix} x_{c_i} \\ y_{c_i} \\ z_{c_i} \end{bmatrix} = 0 \tag{4}$$

which results in

$$\begin{bmatrix} x_{p_1}x_{c_1} & x_{p_1}y_{c_1} & x_{p_1}z_{c_1} & y_{p_1}x_{c_1} & y_{p_1}y_{c_1} & y_{p_1}z_{c_1} & z_{p_1}x_{c_1} & z_{p_1}y_{c_1} & z_{p_1}z_{c_1} \end{bmatrix} E' = 0 \tag{5}$$

where $E' = \begin{bmatrix} E_{11} & E_{12} & E_{13} & E_{21} & E_{22} & E_{23} & E_{31} & E_{32} & E_{33} \end{bmatrix}^T$. Using single value decomposition (SVD), the solution of $E'$ and, consequently, of $E$ are extracted. To calculate the set of inliers we follow the original Libviso2 approach—iterate trough all the matches and use the Sampson Distance [36] to filter the outliers. At the end of all the RANSAC iterations, the final set of matches is available and will be used to compute the essential matrix $E$. Finally, the essential matrix $E$ is computed using this set of matches and imposing the rank-2 constraint to it. Then we extract the camera motion $[R|t]$ from it. However, for a given essential matrix $E$ and considering the previous camera center as reference axis, i.e., $P_p = [I|0]$, there are four different solutions for the current camera matrix $P_c = [R|t]$. To extract the correct solution, a linear triangulation approach is used. Each 3-D unit vector match is triangulated in the following way:

$$\begin{cases} \alpha v_{p_i} = P_p X_i \\ \alpha v_{p_c} = P_c X_i \end{cases} \implies \begin{cases} v_{p_i} \times P_p X_i = 0 \\ v_{c_i} \times P_c X_i = 0 \end{cases} \tag{6}$$

by extending is obtained:

$$\begin{bmatrix} x_{p_i}P_p^3 - z_{p_i}P_p^1 \\ x_{p_i}P_p^2 - y_{p_i}P_p^1 \\ y_{p_i}P_p^3 - z_{p_i}P_p^2 \\ x_{c_i}P_c^3 - z_{c_i}P_c^1 \\ x_{c_i}P_c^2 - y_{c_i}P_c^1 \\ y_{c_i}P_c^3 - z_{c_i}P_c^2 \end{bmatrix} X_i = 0 \tag{7}$$

where $X_i = [x_{t_i} \ y_{t_i} \ z_{t_i}]^T$ and the superscript $P^j$ denotes the $j$-th row of the projection matrix. Solving the linear equation for all the matches considering the four possible $[R|t]$ solutions and choosing the one that presents the higher number of 3-D triangulated points with positive depth results in the final solution for the camera motion. However, from the system of Equation (6) it is visible that the scale factor $\alpha$ was not considered since we use a cross product technique. Thus, the solution for the camera motion $P_c = [R|t]$ is up to a scale factor.

### 4.2. Motion Scale Calculation

To complement the camera motion estimation from our omnidirectional VO approach, a planar LIDAR sensor is considered to recover the scale factor. This approach is divided into four essential steps:

1. Transformation of the range measurement of the LIDAR into the camera referential frame;
2. projection of the LIDAR measures in the camera referential frame into the omnidirectional image;
3. search for associations between image features and LIDAR measures in the omnidirectional image; and
4. scale calculation using the associations found.

To perform the transformation of the range measures to the camera referential frame, we measured the physical distance from the camera center to the LIDAR. We apply a transformation $H = [R|t]$ to the range measures to convert them to the desired referential. The transformation corresponds to

the displacement between the two referentials. For each range measure, we perform the following transformation:

$$\psi_i = H \begin{bmatrix} \kappa_i cos(\theta_i) \\ \kappa_i sin(\theta_i) \\ 0 \end{bmatrix} \tag{8}$$

where $\psi_i = [x_{l_i}\ y_{l_i}\ z_{l_i}]^T$, $\kappa_i$ is the range measure and $\theta_i$ is its correspondent angle. After that, to obtain the range measures as 2-D pixel points in the omnidirectional image, the inverse camera model $M^{-1}$ described in Definition 2 is applied. The final set of 2-D range measures are the ones who are mapped inside the omnidirectional image. In other words, the ones who are inside the camera field of view.

The next step consists in associating the LIDAR measures projected into the image with 2-D feature points present in the current image frame. To do so, a search on the 2-D LIDAR measures neighborhood is computed. As linear searching is computationally expensive, a simplification was performed using the assumption that the vertical standard deviation of LIDAR measures on the image is small. While projecting them into the image, their average $y$ coordinate is computed. In this way, in the first stage, we search for features that present vertical distance to the average smaller than 10 pixels. This allowed us to highly reduce the number of features that are searched in the neighborhood of the LIDAR measures. Then a search is performed in the selected set of features for the ones who present a horizontal pixel distance smaller than five pixels from each LIDAR measure. A representative scheme of this formulation can be found in Figure 5. In Figure 5a is represented the projection of the LIDAR measures in the omnidirectional image viewed from the side. Figure 5b represents the neighborhoods of each projection given by the interception of the yellow zone that represents the first search in $y$ with the blue delimiters that represent the $x$ tolerance of search. The green dots represent image features.

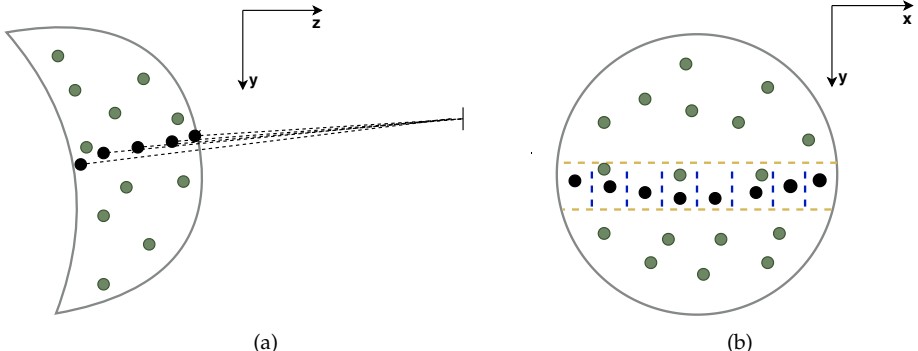

(a) (b)

**Figure 5.** (**a**) Side-view and (**b**) front-view of the projection of light detection and ranging (LIDAR) measures in the omnidirectional image, definition of the neighborhoods and features association.

Figure 6 shows the real projection of LIDAR measures in the omnidirectional image in black and the associated features in white.

A small error is visible associated with the projection due to calibration errors, camera model imperfections, and errors in the projection $H$. This error does not have a high impact on the final estimation of scale due to the filter effect of the performed average described bellow. In other words, the matches between LIDAR measures and 2-D image features that represent a physical outlier, i.e., that do not describe the same point in the world, get diluted in all the other inliers.

After matching 2-D feature points with 2-D range measures, it is possible to estimate the scale factor. These feature points were already triangulated using their respective matches in the previous image frame. Thus, we already have a set of matches between the 2-D LIDAR measures and 3-D triangulated feature points. By consequence, a match is obtained between the raw LIDAR measures

in the world $\psi_i$ and the triangulated feature points $X_i$. This being said, given the set of matches $\{\psi_1, ..., \psi_N\} \longleftrightarrow \{X_1, ..., X_N\}$ the scale factor $s$ is calculated as follows:

$$s = \frac{1}{N} \sum_{i=1}^{N} \frac{||X_i||}{||\psi_i||} \tag{9}$$

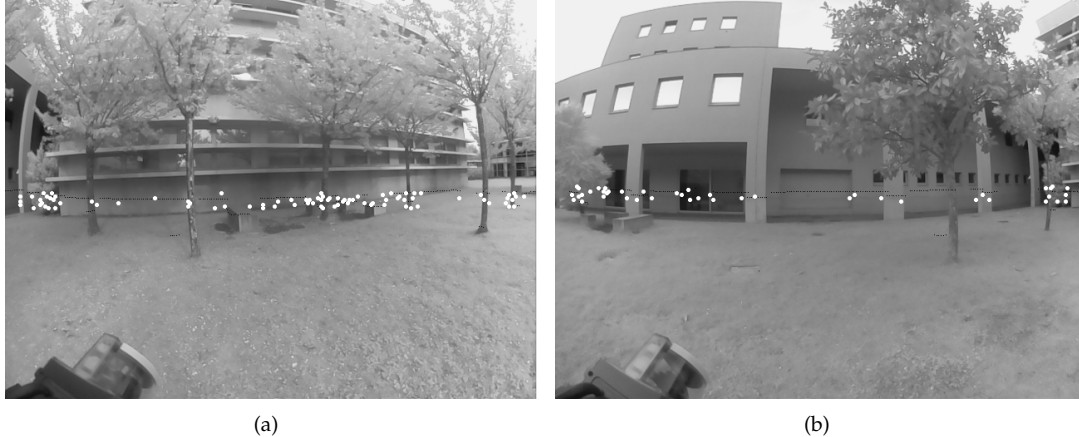

<div align="center">(a)        (b)</div>

**Figure 6.** Two examples (**a**,**b**) of the LIDAR measurements projection in the omnidirectional image and 2-D features association with them.

In other words, the scale factor is the average of the relation between the norm of the triangulated matched features and the distances measured by the LIDAR that are matched. This factor is directly applied to the translation vector extracted from the essential matrix $E$ in the following way:

$$\begin{bmatrix} t_{x_s} \\ t_{y_s} \\ t_{z_s} \end{bmatrix} = \begin{bmatrix} t_x \\ t_y \\ t_z \end{bmatrix} \frac{1}{s} \tag{10}$$

Although in most iterations at least one match between the image features and the LIDAR measures is found, sometimes this does not happen. The last scale factor is used in these cases to prevent the motion from not being scaled.

### 4.3. Orientation Correction

After having a stable camera motion estimation, the need to support it in rotation-only motion types emerged due to high errors in the estimation in these cases. To do so, a gyroscope is used fusing it with the angular velocity resultant from the VO approach using a KF. Besides the angular velocity, the gyroscope bias is also estimated in order improve the motion estimation accuracy. Only the yaw component is estimated because the robot does not present harsh rotations in the other components. Even so, the model is extendable for all the components. So, the state vector is $x = [\omega_x\, \omega_y\, \omega_z\, b_x\, b_y\, b_z]^T$, where $\omega_x, \omega_y, \omega_z$ are the angular velocities states and $b_x, b_y, b_z$ the gyroscope bias. The control vector comes from the VO estimation as $u = [\Delta\theta_x\, \Delta\theta_y\, \Delta\theta_z]^T$ and the observations are $z = [{}^G\omega_x\, {}^G\omega_y\, {}^G\omega_z]^T$ from the gyroscope. This being said, the state model is as follows:

$$\hat{x}_{k+1|k} = A\hat{x}_{k|k} + Bu_k \Leftrightarrow \tag{11}$$

$$\Leftrightarrow \hat{x}_{k+1|k} = \begin{bmatrix} 0 & 0 & 0 & 0 & 0 & 0 \\ 0 & 0 & 0 & 0 & 0 & 0 \\ 0 & 0 & 0 & 0 & 0 & 0 \\ 0 & 0 & 0 & 1 & 0 & 0 \\ 0 & 0 & 0 & 0 & 1 & 0 \\ 0 & 0 & 0 & 0 & 0 & 1 \end{bmatrix} \hat{x}_{k|k} + \begin{bmatrix} \Delta t & 0 & 0 \\ 0 & \Delta t & 0 \\ 0 & 0 & \Delta t \\ 0 & 0 & 0 \\ 0 & 0 & 0 \\ 0 & 0 & 0 \end{bmatrix} u_k \tag{12}$$

witch results in:

$$\hat{\omega}_{i_k} = \Delta \theta_{i_k} \Delta t \tag{13}$$

$$\hat{b}_{i_k} = b_{i_{k-1}} \tag{14}$$

with $i \in \{x, y, z\}$. We considered bias as a constant state ignoring flicker noise and temperature oscillations. Even so, this is a reasonable approximation due to the low impact of these two components in time-limited estimations. In addition the observations model is:

$$\hat{z}_{k+1} = H\hat{x}_{k+1|k} \Leftrightarrow \hat{z}_{k+1} \begin{bmatrix} 1 & 0 & 0 & 1 & 0 & 0 \\ 0 & 1 & 0 & 0 & 1 & 0 \\ 0 & 0 & 1 & 0 & 0 & 1 \end{bmatrix} \hat{x}_{k+1|k} \tag{15}$$

that results, for both components, in an equation of the following form:

$${}^{G}\omega_i = \hat{\omega}_i + \hat{b}_i \tag{16}$$

with $i \in \{x, y, z\}$. This equation can be interpreted as: the angular velocity state is equal to the gyroscope observation minus the bias estimation. So, it is expected that this performs a correction of the angular velocity observation that is used in the computation of the state.

The VO approach in some cases provides unrealistic estimations in pure rotations. Due to this, the covariance matrix of the state $Q$ is dynamic. Both $Q$ and the observations covariance $R$ are initialized with constant values on their diagonals. The values of the diagonal of $Q$ correspondent to the bias states are decreased over time, since bias is considered as a constant state. To detected and cancel the unrealistic peaks of angular velocity on the state, a non-linear approach was adopted. In this, a sigmoid function that varies with the angular velocity states was calibrated. So, if a peak of angular velocity is detected, the sigmoid increases the covariance noise of the angular velocity which leads the filter to consider the gyroscope measure instead.

In this way, it is possible to have in consideration two different sources of information to estimate the robot rotation.

*4.4. Heterogeneous Computing Optimizations*

After having the previously described fusion working on top of a standard computer, we needed it to be fast in an embedded configuration to run on the robot in real-time. So, we chose a low-cost microprocessor—Raspberry Pi 3B—and we tried to optimize the developed code to run on this platform. To do so, we use both Raspberry Pi's CPU and GPU with parallel computing techniques.

To access Raspberry Pi's GPU, the VC4CL (https://github.com/doe300/VC4CL) driver was used. This is an open-source OpenCL 1.2 implementation for Raspberry Pi's GPU that allows the use of OpenCL C++. To facilitate OpenCL usage, an additional layer of abstraction consisting of an OpenCL-wrapper for C++ and ROS was developed. This allowed a communication between the host CPU and the device GPU using simple write and read routines. The implementation layout is represented in Figure 7.

After validating this tool with small kernels like performing arithmetic operations in arrays we moved to the VO algorithm parallelization.

After profiling the VO implementation, we concluded that RANSAC was the most time-consuming block. It runs over 2000 iterations performing several loops and solving equations using SVD in each one. In short, this method is constituted by three main steps:

- randomSample()—calculation of a random set of matches of size 8;
- essentialMatrix()—calculation of the essential matrix *E* using the given set; and,
- getInliers()—calculation of the set of inliers for the given essential matrix *E*.

As the essentialMatrix() routine uses complex calculations it is hard to parallelize in GPU context. We optimize both randomSample() and getInliers() in GPU and essentialMatrix() in CPU.

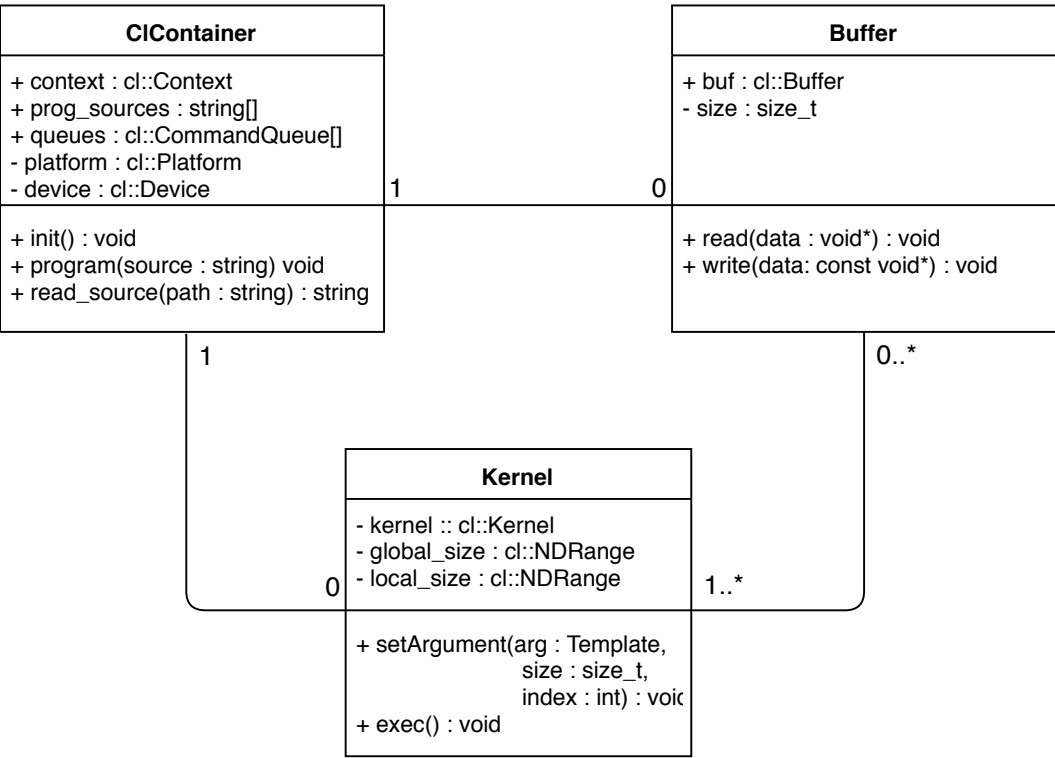

**Figure 7.** Unified modeling language (UML) diagram of the OpenCL abstraction layer implemented.

For the GPU-based optimizations, a 16-way single instruction multiple data (SIMD) kernel architecture was adopted since each quad processing unit (QPU) of this device uses 16-way SIMD, executing an instruction with four-way data parallelism, four cycles in a row. This way, our approach for each routine follows the following pattern:

1. Load the routine input data correspondent to all the RANSAC iterations.
2. Write all the data to the correspondent kernel at once using 16-way vector types.
3. Execute the kernel to all the data in a 16-way vectorized way and load it to a single output array.
4. Read all the output data at once and label the corresponding RANSAC iteration to it.

In this way, we maximize the GPU performance using vectorized kernels that match with its architecture. Also, we minimize the data transfer delays between the host and the device by performing the communication only once for writing and once for reading.

To optimize the essentialMatrix() routine on the CPU, we take advantage of its four cores creating four threads. Each one computes a quarter of all the essential matrices corresponding to a quarter of the total number of RANSAC iterations.

The final solution scheme is represented in Figure 8.

We achieved a parallel RANSAC method that uses both CPU and GPU optimizations.

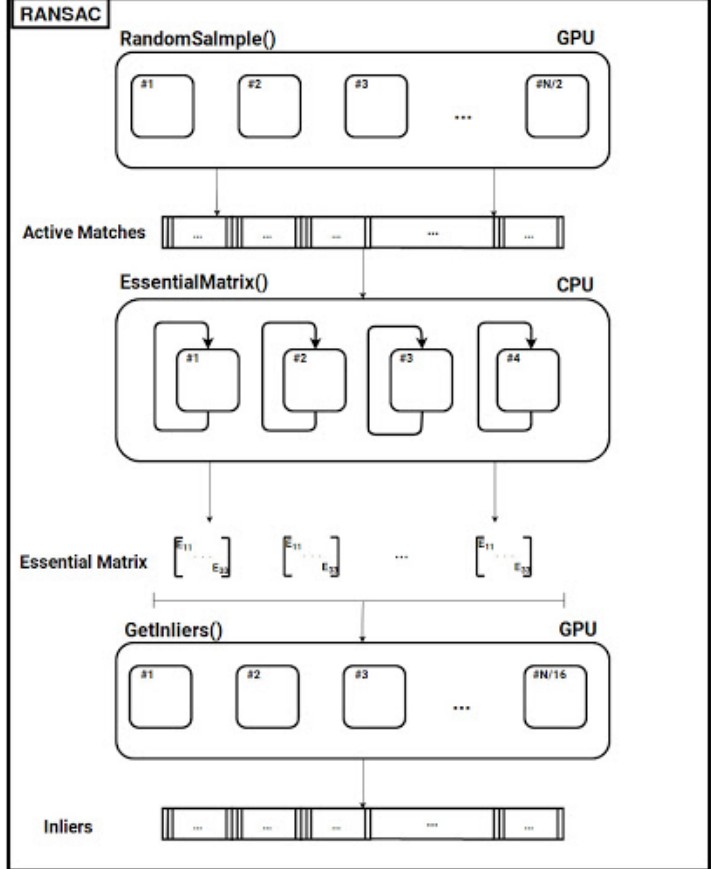

**Figure 8.** Parallel RANSAC configuration.

## 5. Results

To generate the results we built three open-source datasets (http://vcriis01.inesctec.pt/) (with the id DS_AG_38) denoted here as sequences A, B, and C. The first one ran on a computer with the following specifications: Intel(R) Core(TM) i7-4500U CPU @ 1.80GHz and 8 GB RAM. It did not contain parallel computing optimizations. The two other sequences ran on the Raspberry Pi with the developed parallel configuration. All of them were recorded in a garden environment on top of our ground robot. Due to the characteristics of the environment, GPS was not available to generate ground-truth. Thus, in sequences A and B we used Hector SLAM [37] to do so since this method reveals high precision. For sequence C, since the robot performed a small straight trajectory, we used wheel odometry as ground-truth. The vision system was composed of a perspective (https://www.raspberrypi.org/products/camera-module-v2/) and a fisheye (https://www.ptrobotics.com/lcd-cameras-raspberry-pi/5417-raspberry-pi-camera-module-w-fisheye-lens.html) camera to benchmark the original Libviso2 approach with ours. The LIDAR (LMS151-10100) used is the one present in our robotic platform, with the following characteristics: angular resolution of 0.25, an aperture angle of 270° and a scanning range until 50 meters. The gyroscope (UM7 Orientation sensor), also present in our robotic platform, has the following characteristics: EKF estimation rate of 500 Hz, $\pm 1°$ typical static pitch/roll accuracy, $\pm 3°$ typical dynamic pitch/roll accuracy, $\pm 3°$ typical static yaw accuracy, $\pm 5°$ typical dynamic yaw accuracy, 0.5° angle repeatability and 0.01° angular resolution.

### 5.1. Processing Time

To test the performance of the developed parallel RANSAC approach and to analyze its impact on the global processing time of the omnidirectional version of Libviso2, the code was profiled. To do so, the high resolution clock from <chrono> library of std was used. The three optimized functions were

profiled as well as RANSAC itself and the global processing time of the method. Also, to analyze the impact of the parameter that sets the maximum number of features per bucket computed by Libviso2 on the processing time, two values were used for it (Table 1). The two sequences, B and C executed on the Raspberry Pi, were used in this evaluation. This being said, the three optimized methods randomSample(), getInliers, and essentialMatrix were evaluated. Consequently, the RANSAC method itself was also evaluated as well as all the whole FAST-FUSION approach which is denoted as 'process' in Table 1. It is worth to note that the duration times present in this table with respect to the three functions of RANSAC correspond to the time spent by them during all the iterations. Table 1 shows the corresponding results.

**Table 1.** Runtime performance (sec) of the extended version of Libviso2 on Raspberry Pi using both the serial and parallel configurations.

| Sequence | B | | | | C | | | |
|---|---|---|---|---|---|---|---|---|
| Configuration | Serial | | Parallel | | Serial | | Parallel | |
| Features per Bucket | 2 | 3 | 2 | 3 | 2 | 3 | 2 | 3 |
| Random Sample | 0.032 | 0.037 | 0.003 | 0.004 | 0.031 | 0.037 | 0.003 | 0.004 |
| Get Inliers | 0.054 | 0.072 | 0.036 | 0.042 | 0.055 | 0.075 | 0.035 | 0.041 |
| Essential Matrix | 0.255 | 0.251 | 0.095 | 0.096 | 0.247 | 0.249 | 0.094 | 0.094 |
| RANSAC | 0.336 | 0.361 | 0.140 | 0.150 | 0.337 | 0.360 | 0.139 | 0.146 |
| Process | 0.501 | 0.600 | 0.305 | 0.403 | 0.468 | 0.528 | 0.267 | 0.331 |

*5.2. Motion Estimation*

As referenced before, we tested our approach using three different sequences. All of them were placed in an outdoor environment in a ground robot context. In sequence A, we tested the mathematical approach, i.e., the omnidirectional VO system fused with the gyroscope and the laser sensor. The system was executed in a standard computer and benchmarked with the original Libviso2 approach (Figure 9). The two other sequences tested the full system, i.e., the mathematical approach with the heterogeneous computing optimizations and benchmark them with the raw Libviso2 results. Figure 10 shows the results for these two sequences. It is worth noting that we present results with and without the KF that corrects orientation for both the original version of Libviso2 and our FAST-FUSION. This is done to show the importance of this approach and to demonstrate its modularity since it can be coupled to any odometry system.

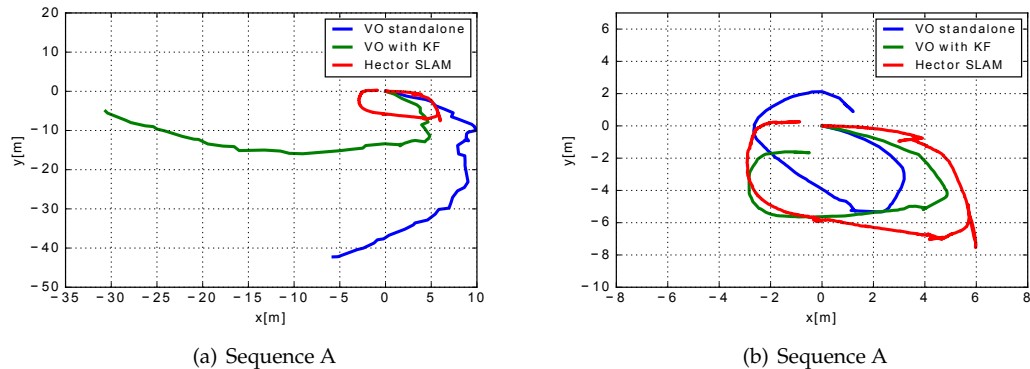

(a) Sequence A　　　　　　　　　　　　　　(b) Sequence A

**Figure 9.** (**a**) Original version of Libviso2 and (**b**) FAST-FUSION motion estimation with and without the orientation correction running on a standard computer, and considering an approximated circle motion.

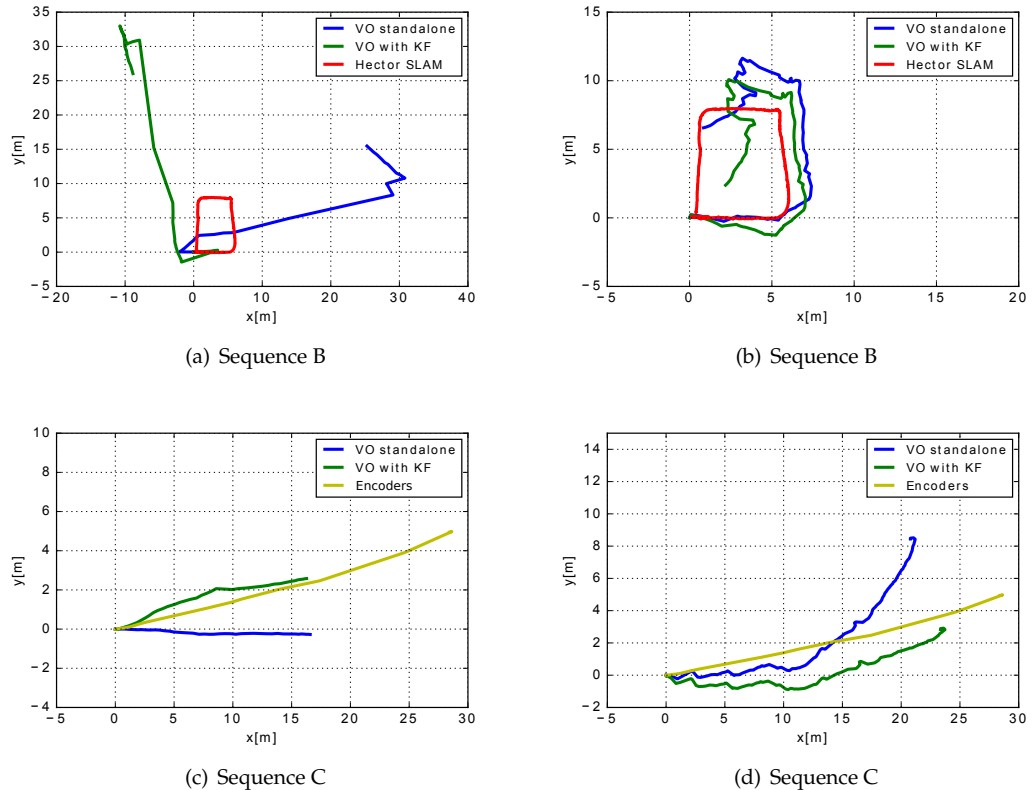

**Figure 10.** (**a**,**c**) Original version of Libviso2 and (**b**,**d**) FAST-FUSION motion estimation with and without the orientation correction running on a Raspberri Pi 3B.

## 6. Discussion

### 6.1. Processing Time

As said before, we tested our parallel configuration on Raspberry Pi 3B using sequences B and C. To analyze the impact of the number of features computed by the VO method on its final runtime performance, two values for it were used, set by the features per bucket parameter.

Starting to look at the RandomSample() method, we can see that this was the least heavy block of RANSAC. It took, on average, about 9.80% of RANSAC's total time for the serial configuration. Using the parallel configuration, we were able to reduce its processing time by 10 times. The GetInliers() method had more weight in RANSAC than the previous one. On average, it took about 18.3% of its total processing time for the serial configuration. It is also visible that the increase in the number of features computed reveals the impact on the performance of this method. This was expected since this method iterated through all the matches in each RANSAC iteration and calculate the Sampson distance to each one. However, for the parallel configuration, this factor did not have the same impact. As the method execution was distributed by the 12 QPUs of Raspberry Pi's GPU, the processing requirements were distributed when increasing the density of features. This led to an increase of gain when increasing the feature density for this method. In fact, using two features per bucket led to an average gain of approximately 1.53 times, and setting this parameter to three resulted in 1.80 times.

By analysis of the EssentialMatrix() method, it is visible that this was the most expensive method of RANSAC. For the serial configuration, it occupied, on average, 72.0% of RANSAC's processing time. This is due to the complexity of SVD calculations and the fact that it is an iterative method. Besides this, the CPU optimization was revealed to be efficient since we obtained an average gain of approximately 2.64 times.

Looking for the final results, we achieved a global average gain of 2.35 times on RANSAC comparing the serial and the parallel configurations. This had a direct impact on the global process runtime performance. Using two features per bucket allowed us to obtain for sequence B an average of 3.28 frames per second and for sequence C 3.75 frames per second. Although for RANSAC the gains were equally good, using three features per bucket resulted in a slower process. For sequence B the result was 2.48 frames per second and for sequence C 3.02 frames per second.

Summing up, we were able to optimize a VO method using heterogeneous computing techniques using a low cost and low power GPU. Using the default level of features per bucket of the original version of Libviso2, which is 2, we achieved embedded real-time performance.

### 6.2. Motion Estimation

The motion estimation accuracy of our approach is based on three main goals:

- Perform a reasonable estimation of the motion scale;
- have the ability to deal with pure rotations; and,
- present real-time performance.

By analysis of Figure 9 is possible to evaluate the two first goals. It is visible that for sequence A, the original version of Libviso2 presented high inaccuracies estimating the motion scale and had difficulties dealing with the rotational component of the robot motion. Applying the KF to the original version improved the estimation of rotation, but globally, the result was still very far from the ground-truth. On the other hand, using a fisheye camera on our FAST-FUSION approach led to a more stable and accurate estimation (Figure 9b). Without the KF, we saw an initial error estimating the rotation that propagates through all the sequence. However, when applying the KF module, this initial error was corrected, and the estimation got closer to the ground-truth.

The final solution results for sequences B and C are present in Figure 10. In this we benchmark the raw Libviso2 version without optimizations with our FAST-FUSION approach. In an embedded system, the impact of computational efficiency was high. Figure 10a proves this. We can see a considerable error in the Libviso2 raw estimation in terms of scale and rotation. In addition, the number of frames that it processes was lower than in our approach since it was slower. This contributed to a degeneration of the motion estimation. For FAST-FUSION, we can see in Figure 10b an estimation much more closer to the ground-truth. Sequence C compared the two VO approaches with wheel odometry in a rectilinear path. Although both approaches revealed error, it is visible that FAST-FUSION estimated a scale factor much closer to the ground-truth (Figure 10d). Consequently, it provided a more accurate estimation.

This being said, we present a version of a state-of-the-art monocular VO method that works with omnidirectional cameras, fused with sensors, and optimized in GPU, providing higher accuracy than the original one. Our system works in an embedded paradigm working in real-time. Besides presenting some estimation errors, we consider the results quite satisfactory, taking into account the conditions: a monocular omnidirectional VO method fused with sensors running in a low-power embedded device.

## 7. Conclusions

We proposed a real-time embedded localization system for ground robots called FAST-FUSION. Our system is composed of an omnidirectional extension of the state-of-the-art monocular VO method Libviso2 that uses raw omnidirectional images, a LIDAR to calculate the motion scale, and a gyroscope to support the estimation in pure rotations. Our VO approach works with raw omnidirectional camera images using a state-of-the-art camera model to consider the lens/mirror distortion. The standalone VO approach is publicly available at the official Libviso2 repository (https://github.com/srv/viso2). We also propose a GPU-based optimization for Raspberry Pi that uses OpenCL to increase the FAST-FUSION frame rate. To test our solution, we created three open-source datasets using our robotic platform. In short, we achieved real-time performance in an embedded configuration,

and our system presented higher accuracy than the original Libviso2 approach. Our system can be implemented considering the following low-cost hardware configuration: a Raspberry Pi 3B (30€), a Raspberry Pi fisheye camera (35.85€), a UM7 Orientation sensor (148.88€), and a RPLIDAR A2M8 360 Degree Laser Scanner (330.00€).

In a future work, instead of using low-level features such as blobs and corners, high-level features, also known as landmarks, will be used. With this evolution a SLAM approach will substitute the current monocular VO method used in FAST-FUSION.

**Author Contributions:** Conceptualization, A.A. and F.B. and A.S.; methodology, A.A. and F.B.; software, A.A.; validation, F.B., A.J.S. and L.S.; formal analysis, F.B. and A.J.S.; investigation, A.A. and F.S. and A.A. and L.S.; resources, F.B. and L.S.; data curation, L.S.; writing–original draft preparation, A.A.; writing–review and editing, A.A. and F.B. and A.J.S.; funding acquisition, F.S.

**Funding:** This research was funded by the ERDF European Regional Development Fund through the Operational Programme for Competitiveness and Internationalisation—COMPETE 2020 under the PORTUGAL 2020 Partnership Agreement, and through the Portuguese National Innovation Agency (ANI) as a part of project "ROMOVI: POCI-01-0247-FEDER-017945". The opinions included in this paper shall be the sole responsibility of their authors. The European Commission and the Authorities of the Programme aren't responsible for the use of information contained therein.

**Acknowledgments:** The authors acknowledge the comments and suggestions from the anonymous reviewers and the assistant editor Amy An for helping improving the paper.

**Conflicts of Interest:** The authors declare no conflict of interest.

## Abbreviations

The following abbreviations are used in this manuscript:

| | |
|---|---|
| VO | Visual odometry |
| SLAM | Simultaneous localization and mapping |
| API | Application programming interface |
| KF | Kalman filter |
| SVD | Single value decomposition |
| CPU | Central processing unit |
| GPU | Graphical processing unit |
| QPU | Quad processing unit |
| ROS | Robot operating system |
| OpenCL | Open computing language |
| GNSS | Global navigation satellite systems |
| LIDAR | Light detection and ranging |
| DSO | Direct sparse odometry |
| SVO | Semi-direct visual odometry |
| SIFT | Scale-invariant feature transform |
| UML | Unified modeling language |

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
