# Peer review of "FAST-FUSION: An Improved Accuracy Omnidirectional Visual Odometry System with Sensor Fusion and GPU Optimization for Embedded Low Cost Hardware†"

_applsci, doi:10.3390/app9245516_

Round 1

Reviewer 1 Report

Well written and detailed. I enjoyed reading it. I only have one suggestion: since the term 'low-cost' was used so many times, would it be possible to mention the prices of those low-cost equipment? You may choose not to use this suggestion. Thank you.

Author Response

Dear reviewer,

First of all I would like to thank you for your reviews.

We will include the following possible configuration in the article.

Raspberry Pi 3B - 30.00€. Raspberry Pi Fisheye Camera - 35.85€. UM7 Orientation Sensor - 148,88€. RPLiDAR A1M8 360 Degree Laser Scanner - 330.00€.

Thank you, once again.

Best regards.

Reviewer 2 Report

Very interesting work, especially from the point wiew of good system and software engineering. Methods are well described and motivated. I would like to have some more information on the hardware side (in particular what kind of LiDAR was used, and its characteristics).

Some points might be made clearer:

1) line 161, def. 4.1: Please clarify what camera plane and sensor plane mean. I assume this also related to other currently used expressions such as focal and image plane.

2) line 164: Is it possible to estimate depth from a static image just from r? r also depends on the angle of view... I assume that obviously the answer is yes, since your method does work, but maybe a few words more would make the concepts clearer...

3) line 212 (about Kalman filtering): How are you initializing, weighing, and updating variance matrices?

4) line 261: Please give some information on the LiDAR used: technology (is it a solid-state LiDAR?), characteristics (number of segments, or points per second, FOV, range, expected accuracy...). Maybe also info on the gyro.

A few small notes:

line 24: ...difficult the robot motion estimation... -> ...make robot motion estimation difficult...

line 31: preferably define acronym VO in the text before using it (e.g. at line 13) (even if there is an acronym table in appendix)

line 66: In the next section is presented the related work. -> In the next section the related work is presented.

line 74 ... approaches solve... -> ... approaches to solve...

line 193: Then is searched in the selected set of features for the ones... -> Then a search is performed in the selected set of features for the ones...

Author Response

Dear reviewer,

First of all I would like to thank you for your relevant comments and reviews.

Follows our answers to all of them.

We used the nomenclature provided by the authors of the camera calibration model/toolbox. The camera plane refers to the image plane and it is expressed in pixels. The sensor plane is an hypothetical plane orthogonal to the mirror axis or fisheye lens, with the origin located at the camera optical center. It is expressed in metric coordinates, i.e., pixels in relation with the image center. We will include this information into the camera model section. The camera model allows us to convert pixels present in the omnidirectional image into 3-D unit vectors, as referenced in the article. In other words, we can extract the direction of the ray that passes through the 3-D world point correspondent to the image pixel. So, your question is extremely valid. That sentence is confuse and we should use the world direction instead of depth.We will correct this miss understanding. This is actually a relevant part of our work that we chose to omit, but we will now insert a brief description of it in the article. The covariance of the observations is initialized with fixed values on its diagonal and they are conserved during the process. The covariance of the state is also initialized with constants on its diagonal but it is dynamic. Since we consider the gyroscope bias as a constant state we decrease its covariance in each iteration. In what concerns with the angular velocity states we use a non-linear approach to update its covariance. Since we obtain sometimes unrealistic estimations during pure rotations (peaks of angular velocity) from Visual Odometry, if we consider them the bias estimation will degenerate as well as the final motion estimation. So, to cancel the Visual Odometry noisy estimations in pure rotations we calibrated a sigmoid function that varies with it. So, if a peak of angular velocity is detected the sigmoid makes the covariance error of the angular velocity states very high which makes the filter consider the gyroscope measure instead.We will describe this procedure briefly in the Kalman Filter section. Besides our system does not require a LIDAR with this characteristics, we used the laser that is already present in our robotic platform for convenience. It as angular resolution of 0.25, an aperture angle of 270 and a scanning range until 50 meters. In the same way, our robot has the gyroscope that is present in the UM7 Orientation Sensor.We will provide a full possible configuration to work with our system in the article -  Raspberry Pi 3B, Raspberry Pi Fisheye Camera, RPLiDAR A2M8 360º Laser Scanner, UM7 Orientation Sensor.

We will also make the changes relative with the English notes.

Thank you, once again.

Best regards.